# An Albumin Biopassive Polyallylamine Film with Improved Blood Compatibility for Metal Devices

**DOI:** 10.3390/polym11040734

**Published:** 2019-04-23

**Authors:** Shuang Lin, Xin Li, Kebing Wang, Tengda Shang, Lei Zhou, Lu Zhang, Jin Wang, Nan Huang

**Affiliations:** Key Lab. of Advanced Technology for Materials of Education Ministry, School of Materials Science and Engineering, Southwest Jiaotong University, Chengdu 610031, Sichuan, China; ls1058030562@163.com (S.L.); wang12917@126.com (K.W.); tengdashang182156@163.com (T.S.); xnjtdxzhoulei@163.com (L.Z.); 13920365680@163.com (L.Z.); huangnan1956@163.com (N.H.)

**Keywords:** plasma polymerization, blood-contact devices, polyallylamine, BSA, blood compatibility

## Abstract

Nowadays, a variety of materials are employed to make numerous medical devices, including metals, polymers, ceramics, and others. Blood-contact devices are one of the major classes of these medical devices, and they have been widely applied in clinical settings. Blood-contact devices usually need to have good mechanical properties to maintain clinical performance. Metal materials are one desirable candidate to fabricate blood-contact devices due to their excellent mechanical properties and machinability, although the blood compatibility of existing blood-contact devices is better than other medical devices, such as artificial joints and artificial crystals. However, blood coagulation still occurs when these devices are used in clinical settings. Therefore, it is necessary to develop a new generation of blood-contact devices with fewer complications, and the key factor is to develop novel biomaterials with good blood compatibility. In this work, one albumin biopassive polyallylamine film was successfully established onto the 316L stainless steel (SS) surface. The polyallylamine film was prepared by plasma polymerization in the vacuum chamber, and then polyallylamine film was annealed at 150 °C for 1 h. The chemical compositions of the plasma polymerized polyallylamine film (PPAa) and the annealed polyallylamine film (HT-PPAa) were characterized by Fourier transform infrared spectrum (FTIR). Then, the wettability, surface topography, and thickness of the PPAa and HT-PPAa were also evaluated. HT-PPAa showed increased stability when compared with PPAa film. The major amino groups remained on the surface of HT-PPAa after annealing, indicating that this could be a good platform for numerous molecules’ immobilization. Subsequently, the bovine serum albumin (BSA) was immobilized onto the HT-PPAa surface. The successful introduction of the BSA was confirmed by the FTIR and XPS detections. The blood compatibility of these modified films was evaluated by platelets adhesion and activation assays. The number of the platelets that adhered on BSA-modified HT-PPAa film was significantly decreased, and the activation degree of the adhered platelets was also decreased. These data revealed that the blood compatibility of the polyallylamine film was improved after BSA immobilized. This work provides a facile and effective approach to develop novel surface treatment for new-generation blood-contact devices with improved hemocompatibility.

## 1. Introduction

Large quantities of blood-contacting medical devices are used annually world-wide. When an artificial material comes into contact with blood, coagulation and thrombus formation immediately occur [1]. Thus, blood compatibility and antithrombogenicity are the most important properties required for blood-contacting medical devices. Hence, antithrombogenic treatments for blood-contacting devices were developed in the second half of the 20th century [2]. However, none of the blood-contacting materials used in clinical settings can meet all the hemocompatibility requirements, whether short-term applied or the long-term implanted. These devices include blood purification devices, vascular stents, catheters, artificial heart valves, inferior vena caval filter, sensing devices, and so on [3,4,5,6,7]. These devices urgently require preferable hemocompatibility to maintain device performance.

According to their mechanical properties, metals are used to construct medical devices. The metals can be used in pure form or combined with other elements (metal alloys), for instance, titanium (Ti), Ti-based alloys, stainless steel, Co-based alloys, and Mg and Mg-based alloys [8,9,10,11]. One major application of these materials is fabricating blood-contacting devices, such as coronary, inferior vena caval filter, vascular, and biliary stents [12,13]. Thrombus formation is one of the key blood-contact complications for these devices in vascular applications. Hence, numerous strategies were developed to modify these metal devices for preventing thrombus formation. Surface modification strategies enable the combination of a material’s bulk properties with desired biological attributes, and it has become a focus strategy for preventing cruor. This surface modification strategy could be realized in many ways, for instance, surface biopassive, surface bioactive, and biomimetic modification [14,15,16]. One of the most well-known methods is developing heparin-grafted materials. A series of new heparinized films were fabricated by surface-grafting process on polymers, such as chitosan/soy protein films and expanded PTFE films [17,18]. In addition, bivalirudin, a synthetic anticoagulant that is a 20-peptide analog of hirudin, is used for surface modification. Chen et al. immobilized bivalirudin (BVLD) in an organic phytic acid (PA) coating on Mg by an in situ chemical route; the results showed a prolonged clotting time, inhibited platelets adhesion, as well as reduced hemolysis compared to untreated Mg. Hirudin is a natural bio-molecular inhibitor that functions as a serine proteinase to efficiently prevent blood coagulation [19]. Li et al. immobilized the natural hirudin on polyvinylpyrrolidone (PVP) crosslinking membrane via hydrogen bonding, which shows desired anti-clotting stability and activity [20]. Enzymes are also introduced into the coating to fabricate surface bioactive surface. Xue et al. reported the design of a graphene–haemin–glucose oxidase conjugate as a tandem catalyst; enzymatic catalyst glucose oxidase was used for biomimetic generation of antithrombotic species. The conjugates enable the generation of nitroxyl, an antithrombotic species, from physiologically abundant glucose and L-arginine. The conjugates could be embedded within polyurethane to create antithrombotic coating for blood-contacting devices [21].

The surface biopassive mothed could make the surface of the biomaterials induce minimal interactions with the blood, and it has proved popular. There are many compounds used to constructed the biopassive surfaces, such as polymers and biomolecules. Various brush-forming polymers are applied to develop the passive surfaces, such as polyethylene glycol (PEG) [22], poly (ethylene) oxide (PEO) [23], and polyhydroxyethyl methacrylate (HEMA) [24]. Some zwitterionic molecules are also used to modify the blood-contact materials surfaces, for instance, phosphorylcholine methacrylate (MPC) [25] and glycine betaine [26]. Biomolecules, such as proteins, could also be used to fabricate the biological passive surface, especially the albumin. Surface coating with albumin as a passivation strategy has been intensively studied since 1986 [27,28], and it has proved popular. Albumin pre-adsorption treated surfaces could inhibit the adhesion of platelets and leukocytes, and it also could prevent the activation of enzymes of the coagulation cascades [29,30,31]. However, the pre-adsorbed albumin could be easily displaced by other proteins, quickly hampering performance. These molecules-modified materials both reveal adhesion resistance to some extent. However, it is difficult to introduce these materials onto the surface of metal devices directly. In this work, a polyalanine platform was created to immobilize albumin in a mild condition. The albumin was fixed onto the surface by chemical reaction. This albumin-immobilized film was more stable and durable.

In this work, the polyallylamine film was prepared via pulsed plasma, and then annealed under vacuum. This film provided a good platform for albumin immobilization. The surface chemical composition was characterized by FTIR and XPS. The surface topography was evaluated by AFM, and the surface wettability was evaluated by wetting angle. The stability of these polyallylamine films were also tested. Later then, the bovine serum albumin (BSA) was immobilized onto the polyallylamine surface, and FTIR and XPS were employed to confirm the successful introduction of albumin. The numbers and activation of the platelets adhered on the surface of albumin-modified polyallylamine film markedly decreased. Our work provides one approach to building biological passive film on the metal biomaterials surface for improved blood compatibility.

## 2. Materials and Methods

### 2.1. Materials

Allylamine was obtained from Sigma-Aldrich (Saint Louis, MO, USA). LDH assay was provided by Sigma-Aldrich. BSA were provided by Amresco (Solon, OH, USA). The 316L stainless steels were purchased from Baowu (Shanghai, China). Glutaraldehyde aqueous solution (30%) and ethanol were obtained from Keshi (Chengdu, China). Phosphate-buffered saline tablet was provided by Sigma-Aldrich. Other reagents were local products of analytical grade.

### 2.2. Methods

#### 2.2.1. Plasma Polymerization of the Polyallylamine Films

The polyallylamine film was deposited on the mirror polished 316L stainless steel (SS) by plasma polymerization system. The system was made by SP-II type radio frequency (RF) matcher, 500 W pulsed RF power, and the vacuum reaction chamber. The deposition conditions were as follows: pressure 10 Pa, flow rate of Ar 4.55 sccm, pulse duty factor 40%, direct current bias −75 V, RF power 120 W, and deposition time 45 min. The samples were named as PPAa. Later, the PPAa was annealed at 150 °C under ≤1.0 × 10^3^ Pa for 1 h, and the annealed PPAa film was named as HT-PPAa.

#### 2.2.2. Surface Characterization of the Polyallylamine Films

The surface chemical compositions of the samples were measured by X-ray photoelectron spectroscopy (XPS, XSAM800, Kratos Ltd., Manchester, UK, Al Ka). The resolution was 1 eV for the wide scans and 0.1 eV for the high-resolution tests. The take-off angle was 90°. The binding energies were calibrated using C 1s and charge-compensated using an electron neutralizer. The chemical structure of the PPAa and HT-PPAa films was analyzed by Fourier transform infrared spectroscopy (FT-IR, NICOLET 5700, Thermo Fisher, Waltham, MA, USA). The surface morphology and roughness of the films were detected by an atomic force microscope (AFM). The surface contact angle of the films with DI water was measured by DSA100 (Krüss, Hamburg, Germany) using the sessile drop method at room temperature.

#### 2.2.3. Film Stability

The stability of the films was also measured. The samples were immersed into the phosphate-buffered saline (PBS) (pH 7.4) at 37 °C under dynamic conditions. After 30 days immersion, the samples were taken out and observed using scanning electron microscopy (SEM, Quanta 200, FEI, The Netherlands).

#### 2.2.4. BSA Immobilization and Characterization

The annealed PPAa films were immersed into the glutaraldehyde solution (1 wt %) for 3 h at room temperature. After being washed with DI water, the films were immersed into the BSA solution (10 mg/mL, 4 °C, 24 h). Subsequently, the films were washed three times using DI water, and these films were named as BSA-HT-PPAa. The chemical structure and composition of the BSA immobilized film were determined by FTIR and XPS, respectively. Meanwhile, the surface contact angle of the modified film with DI water was also measured by DSA100 (Krüss, Hamburg, Germany) using same methods.

#### 2.2.5. In vitro Hemocompatibility Evaluation

For the blood compatibility evaluation, the platelet rich plasma (PRP) was obtained by centrifuging (1500 rpm, 15 min) using fresh collected blood. The fresh PRP was dropped onto the surface of each sample and incubated for 2 h at 37 °C. After washing with PBS solution, the samples were fixed with 0.5% glutaraldehyde solution for 12 h.

After being washed with PBS three times, the samples were dehydrated by immersing in 50, 75, 90, and 100% ethanol solutions for 10 min, respectively. After critical point drying, the samples were sputtered with gold and examined by SEM. The amount of adhesion platelets was determined by the lactate dehydrogenase (LDH) assay and the detailed operation procedures referred to in our previous works [32].

### 2.3. Statistical Analysis

The data were statistically evaluated using ANOVA via the homogeneity test of variances. Subsequently, a post-hoc test was performed using the LSD method for comparison. The data were expressed as mean ± standard deviation (SD). The probability value was considered as a significant difference when *p* < 0.05. The data were analyzed using SPSS 11.5 (Chicago, IL, USA).

## 3. Results and Discussion

The chemical structures of the PPAa and the annealed PPAa films were characterized by FTIR. As shown in Figure 1, the relative absorptions of the C-H (3092 cm^−1^) and C=C (1631 cm^−1^, 920 cm^−1^) from the -CH=CH_2_ groups were observed according to the data of allylamine monomer and according to the absorption peaks of the CH_2_ groups (2924 cm^−1^, 2857 cm^−1^), -NH_2_ groups (1575 cm^−1^) [33]. For the PPAa film, the peaks at 920 cm^−1^ and 3092 cm^−1^ belong to the disappeared -CH=CH_2_ groups, and the peaks at 1631 cm^−1^ and 1575 cm^−1^ were both blue-shifted. The plasma opened the C=C bond of the allylamine, and then induced the addition polymerization [33]. Hence, the absorption peaks of -CH=CH_2_ groups disappeared. Meanwhile, the primary amine groups on the monomer were able to be conversed to the -C=NH and nitrile groups during the polymerization process. So, the peak at 1631 cm^−1^ shifted to 1651 cm^−1^ (C=N), and the peak at 1575 shifted to 1591 cm^−1^ (C-H in C=N groups). The weak absorption in 2200 cm^−1^ was contributed to by the nitrile groups. After 150 °C annealed, the relative intensity of peak at 1651 cm^−1^ was increased in the HT-PPAa film. The heat treatment could induce the polyimide type structures according to the literature [34], and it could form more C=N structures. The FTIR data confirmed the successful preparation of PPAa and the polymerization of the allylamine induced by plasma. The annealing process could further increase polymerization.

The surface topography and roughness of the PPAa and HT-PPAa films were measured by AFM. The surfaces of the PPAa and HT-PPAa both presented island structures, as shown in Figure 2, and the island structures were smaller after being heat treated. The roughness of the HT-PPAa film (0.298 nm) also decreased compared with the PPAa film (0.381 nm); this may have been due to the further polymerization and the volatilization of the low polymerization polyallylamine during the annealing process. The cross-linking reactions could have happened in the polyalanine film during the annealing process, and further cross-linking could also have caused a slight contraction. The contraction could have caused the slight stripes to arise, which is apparent in the AFM data of the HT-PPAa film.

The wettability of the biomaterials was crucial for their biocompatibility, including their hemocompatibility. The water contact angle of PPAa surface was lower than SS surface, as shown in Figure 3, and the water contact angle of HT-PPAa was similar to the untreated film. This data indicated that the wettability was increased by the introduction of the PPAa film. Additionally, the thickness of the polyallylamine films was also evaluated, as shown in Figure 4. The thickness of the PPAa film was about 105 nm, and it decreased to 82 nm after heat treatment. This could have been due to the volatilization of oligomeric polyallylamine and the further cross-linking reactions during the annealing treatment.

As shown in Figure 5, the surface -NH_2_ density of the PPAa was about 53.6 nmol/cm^2^. After heat-treatment, the density of the -NH_2_ on HT-PPAa surface was reduced to 45.3 nmol/cm^2^. The loss of the primary amino group may have been due to the thermal evaporation of the oligomer on the surface; in addition, the further crosslinking of PPAa at a high temperature to form polyimide structure may have also consumed a portion of primary amino groups. In general, the presence of an amount of surface -NH_2_ makes it possible to graft biomolecules.

The stability of the films was a key factor for their application, regardless of the short-term or long-term benefits. Hence, the PPAa film and the HT-PPAa film were immersed into the PBS solution for 30 days to evaluate the stability of these films, as shown in Figure 6. Before immersion, the surfaces of these two films were flat and uniform and there were no significant defects. The obvious cracks were observed on the surface of PPAa film, after 1-day immersion. There were only some stripes observed on the surface of the HT-PPAa film. After 10 days immersion, the number of cracks were increased on the PPAa surface and the HT-PPAa film surface presented more stripes. However, no cracking occurred on the HT-PPAa film. This difference may due to the cross-linking reactions in the PPAa film after annealing; further damage was limited by the transversal interactions between chemical groups. Moreover, the stripping and cracking were both observed on the PPAa film after 30 days immersion, while cracking was also observed on the HT-PPAa film. This may due to the partical cross-linking of the HT-PPAa film. The immersion data indicated that the HT-PPAa film had better stability, which may due to the further cross-linking after heat treatment. The HT-PPAa film was used to modify the blood-contacting devices, and it maintained the properties for at least 10 days.

There were a lot of amino groups on the surface of HT-PPAa film, which could have been used to immobilize a large number of molecules. The BSA was immobilized onto the HT-PPAa surface to build biological passive surface, looking forward to the development of hemocompatibility materials. XPS analysis of the HT-PPAa film and BAS-modified film was carried out. High-resolution C (1s) XPS spectras of two films are shown in Figure 7. The broad peak of the C1s was resolved into two parts, C-C/C-H (284.8 eV) and C-N/C=N/C≡N (286.2 eV), as shown in Figure 7A, and this result was consistent with the FTIR data. The C=C bonds in the allylamine monomer were opened by the plasma and then polymerized to C-C bonds, and heat treatment induced further polymerization. The C-C bonds were the main structural component of the HT-PPAa. After being BSA-modified, the shape of the peak was changed. The intensity of the peak at high bonding energy was increased, due to the introduction of C=O bonds from abundant CONH bonds in BSA. Meanwhile, the content of the peak at 285.9 eV was increased. The XPS data verified that the BSA was successfully immobilized onto the surface of the HT-PPAa film.

The surface chemical structures of the BSA-modified film were analyzed by FTIR, as shown in Figure 8. For the BSA-HT-PPAa film, absorption of the peak at 3300 cm^−1^ was increased due to the N-H groups in the BSA molecule. The broad peak at 1651 cm^−1^ in the HT-PPAa turned into two peaks (1690 cm^−1^ and 1580 cm^−1^) after being BSA-modified. This due to the obsoption of amide I band (1685 cm^−1^) and amide II band (1550 cm^−1^) in the abundant CONH bonds on BSA-modified film. The FTIR data also confirmed the successful immobilization of BSA onto the HT-PPAa surface.

The surface wettability was also evaluated after BSA immobilized. The water contact angle of the BSA-HT-PPAa surface was decreased compared with the surface of the HT-PPAa film, as shown in Figure 9. There were a lot of hydrophilic groups in the BSA molecules, such as –NH_2_, –OH, and –COOH [35]. Hence, the introduction of BSA increased the hydrophilicity of the surface.

The blood-contacting devices could lead to unwanted coagulation and immune response when applied to the blood environment [36]. The surface treatment aimed to get better hemocompatibility. Platelets behavior was a key point in the coagulation process [37]. Hence, the platelets adhesion and activation behavior on surface of the polyallylamine films were researched in this work. The platelets adhered on the SS surface presented multiple forms, and some of the platelets fully spread out. A large number of the adhered platelets aggregated, and the pseudopodia also extended. The aggregated platelets were also observed on the PPAa and HT-PPAa films surfaces, and the number of the adhered platelets on these two films surfaces was more than that on SS surface. There were no aggregated and spreaded platelets observed on the BSA-modified surface. The number of the adhered platelets on the BSA-HT-PPA surface was reduced significantly. The LDH assay was also employed to test the platelet adhesion ratio, the results were presented in Figure 10B. the platelets adhesion rates on the PPAa film and the HT-PPAa film were higher than that on the SS surface, and that may due to the plenty of amino groups induced platelet activation on polyallylamine films surfaces. After being BSA-modified, the surface platelets adhesion rate of the BSA-HT-PPAa was much lower than that in the un-modified films. BSA is negatively charged, and the BSA-modified surfaces were also negative charged. Negatively charged surfaces can inhibit the adhesion of red cells, platelets, cells, and immune cells, which are also negatively charged. Besides, albumin-modified surfaces are short of peptide sequences for interactions with platelets, leukocytes, and enzymes of the coagulation cascades. These platelets adhesion data indirectly verified the improved blood compatibility of the BSA-HT-PPAa film. In general, this BSA-modified polyallylamine film could be constructed on the metal material surface, and the coated metal materials could obtain favorable blood compatibility.

## 4. Conclusions

In this work, one kind of polyallylamine film was constructed by plasma polymerization onto biomedical stainless-steel surface, and then this film was annealed. The annealed film had acceptable stability within 10 days, and it satisfied the requirements of most short-term blood-contact devices. BSA biological passive surface was successfully established on this polyallylamine film. This BSA biological passive film revealed good blood compatibility. This BSA functionalized polyalanine film was able to be constructed onto the surfaces of multiple materials, because this film formation was not limited by the substrates. Hence, it was easily coated onto the surfaces of metals. Two limitations of the metals—that they were difficult to modify and had limited hemocompatibility—were able to be solved at the same time. Furthermore, this polyalanine film was also able to be used to immobilize a variety of molecules that contained amino groups, carboxylic group, and double bond. This work provides one strategy for blood-contact metal devices to improve their blood compatibility, prevent clotting, and achieve better clinical performance. This work also offers one platform and one methed to immobilize functional molecules onto bioinert materials surfaces.

## Figures and Tables

**Figure 1 polymers-11-00734-f001:**
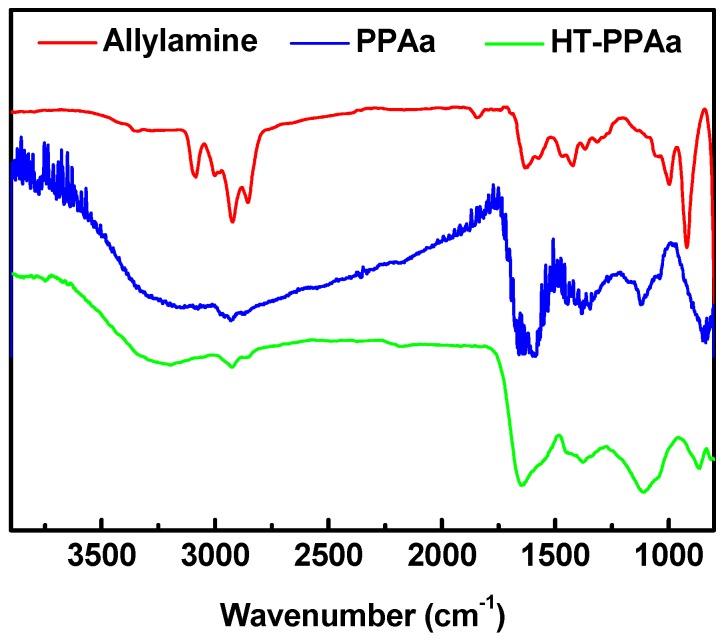
Fourier transform infrared spectrum (FTIR) results of allylamine, plasma polymerized polyallylamine film (PPAa), and annealed polyallylamine film (HT-PPAa) film.

**Figure 2 polymers-11-00734-f002:**
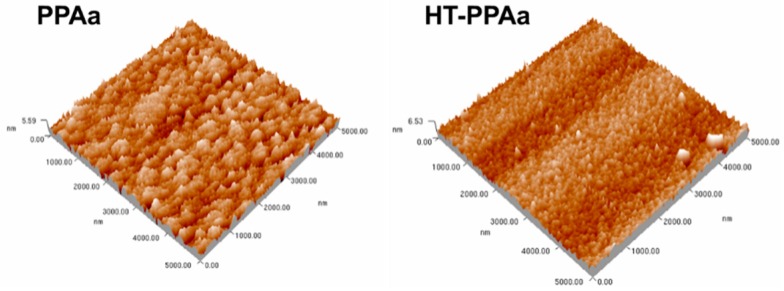
Atomic Force Microscopy (AFM) results of the PPAa and HT-PPAa film surface topography.

**Figure 3 polymers-11-00734-f003:**
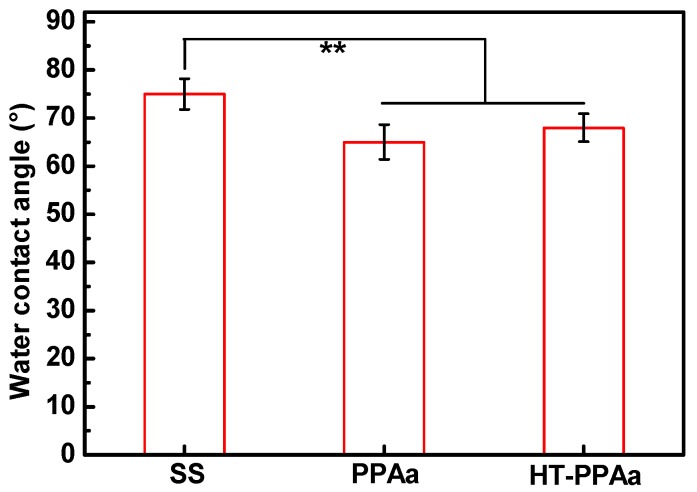
The water contact angle of stainless steel (SS), PPAa, and HT-PPAa film. Data are presented as mean ± SD (*n* = 4) and analyzed using one-way ANOVA (* *p* < 0.05, ** *p* < 0.01, and *** *p* < 0.001).

**Figure 4 polymers-11-00734-f004:**
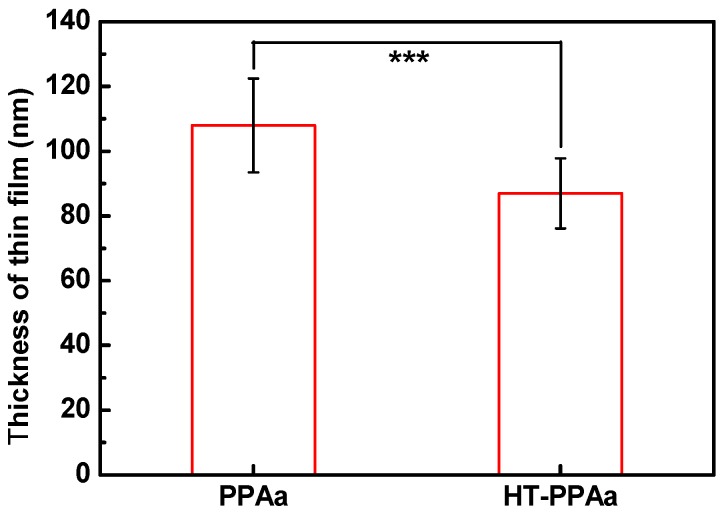
The thickness of PPAa and HT-PPAa film. Data are presented as mean ± SD (*n* = 4) and analyzed using one-way ANOVA (* *p* < 0.05, ** *p* < 0.01, and *** *p* < 0.001).

**Figure 5 polymers-11-00734-f005:**
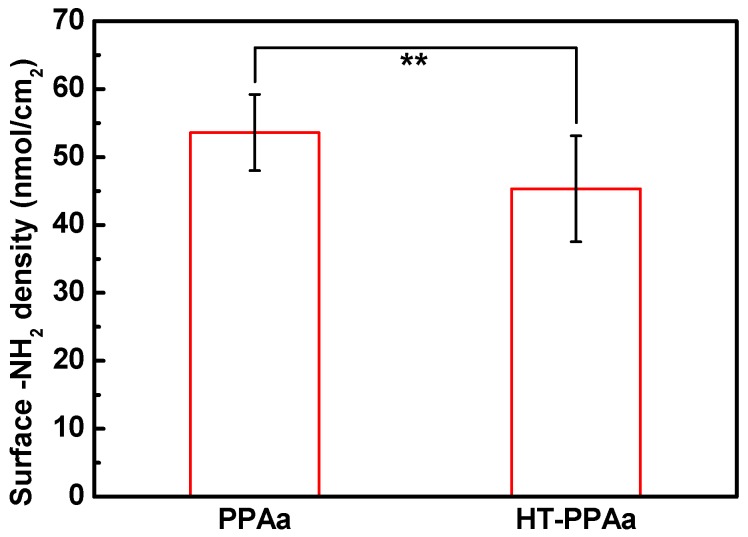
Surface -NH_2_ density of the PPAa and HT-PPAa samples. Data are presented as mean ± SD (*n* = 4) and analyzed using one-way ANOVA (* *p* < 0.05, ** *p* < 0.01, *** *p* < 0.001).

**Figure 6 polymers-11-00734-f006:**
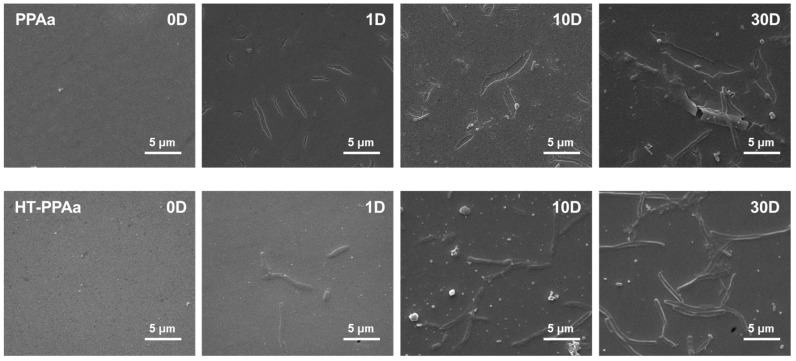
The SEM images of PPAa film and HT-PPAa film immersed into the phosphate-buffered saline (PBS) solution for 30 days.

**Figure 7 polymers-11-00734-f007:**
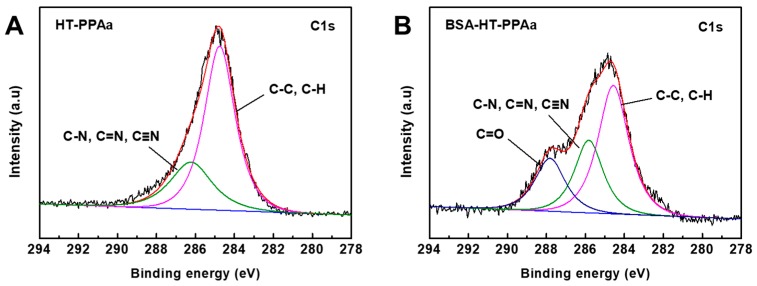
High-resolution survey of (**A**) HT-PPAa film and (**B**) BSA-HT-PPAa film on C1s.

**Figure 8 polymers-11-00734-f008:**
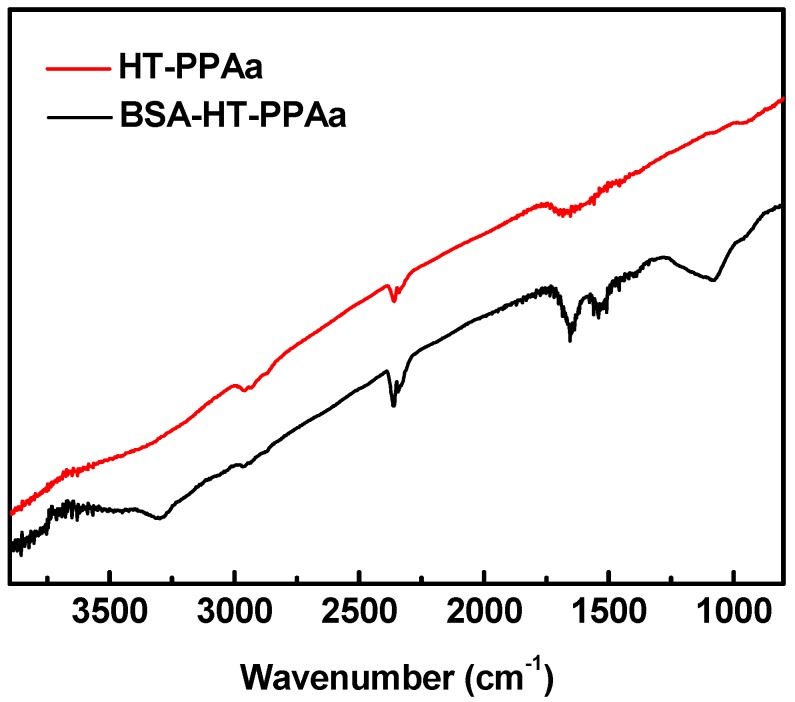
FTIR results of HT-PPAa film and BSA-HT-PPAa film.

**Figure 9 polymers-11-00734-f009:**
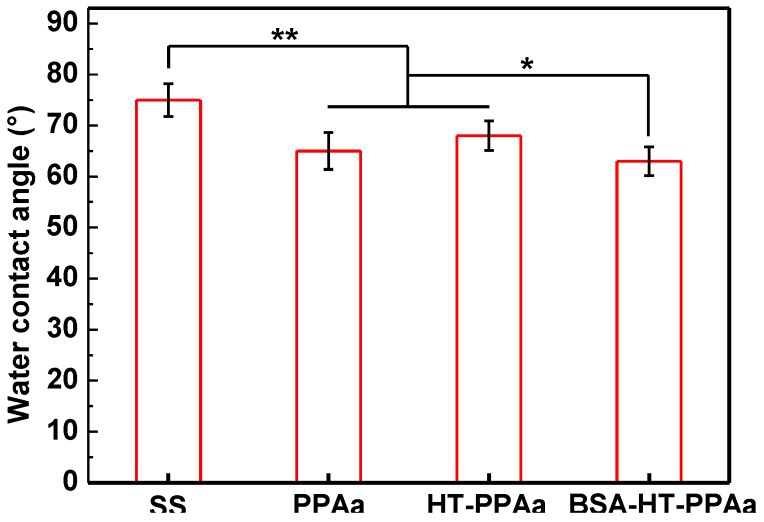
The water contact angle of SS, PPAa, and HT-PPAa and BSA-HT-PPAa film. Data are presented as mean ± SD (*n* = 4) and analyzed using one-way ANOVA (* *p* < 0.05, ** *p* < 0.01, and *** *p* < 0.001).

**Figure 10 polymers-11-00734-f010:**
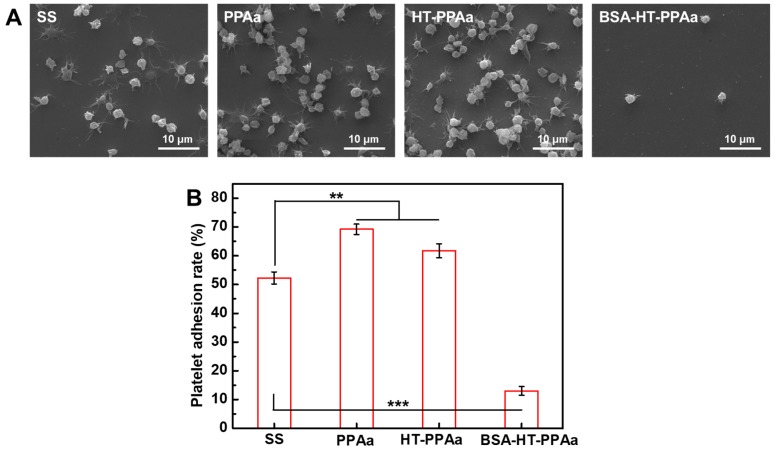
(**A**) The platelet adhesion rate of SS, PPAa, HT-PPAa, and BSA-HT-PPAa film. (**B**) Data are presented as mean ± SD (*n* = 4) and analyzed using one-way ANOVA (* *p* < 0.05, ** *p* < 0.01, and *** *p* < 0.001).

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
