# Peer review of "An Albumin Biopassive Polyallylamine Film with Improved Blood Compatibility for Metal Devices"

_polymers, 2019, doi:10.3390/polym11040734_

Reviewer 1 Report

Dear authors

First of all, let me congratulate you for the work developed in this paper.I only will suggest some light changes, just with the aim to improve, if possible, the understanding of readers .

, Abstract: lines 16-19, please consider to revise the text, I think that there're some mistakes

- Introduction: Lines 56 and 59, please, consider also the revison of the text. Specially, line 59 when talking about "cascade of reactions" . For to accomplish coagulation. Are there some reactions involved?? and which are the type of reactions?? When you check results, you take, only, into account the possibility of coagulation, but the presence of NH2 groups on surface (as you demostrated clearly) will allow some secondary reactions

- Line 141. You refer to an especific structure of polyimide that is the same as "according to literature". I've searched on internet and there are several possibilities for this structure. It would be interessting to add some bibliographic references to make easy to understand the trascendency of this "structure"

-Line 161: You attribute several physical changes on the surface of HT-PPA's to the loss of colatile compounds or low molecular weight systems. This can explain some results, but, in my opinion, in a system that has been activated by plasma treatment, when annealing there shuold be some cross-linking reactions attending to the chemical structures involved and their tridimensional orientation. That fact should promotte a light contraction that would explain holes and strives that can be seen on Fig2 and Fig 6. Those bonds can not be detected using ATR.FTIR

-Line 164. In my opinion, and from data in graph on Fig 5, density of PPAa should be more 54 or 53 than 45.3 nmol/cm2. The expression used "reduced" used after, confirms my impresion

-Line 175-176.Swelling, usually never creates cracks or stripes, because is limited by the transversal interactions between chemical groups, ant it is usually limited except in viscoelastic systems. In my opinion, that will be caused by partial cross-linking effects, mentioned before.

Thanks a lot for your work, very interessting

Author Response

Thank you for your letter and the reviewers’ comments concerning our manuscript entitled “An albumin biopassive polyallylamine film with improved blood compatibility for metal devices(polymers-486496). Those comments are all valuable and very helpful for revising and improving our paper, as well as the important guiding to our researches. We have studied reviewer’s comments carefully and made revision which marked in blue in the text. We have tried our best to revise our manuscript according to the comments, which we hope to meet with approval.

The list of revision and reply:

1 CommentAbstract: lines 16-19, please consider to revise the text.

Response: Thanks for the Reviewer’s comments. Lines 16-19 were revised to “Therefore, it is eagerly to develop new generation of blood contact devices with less complications, and the key factor is to develop novel biomaterials with good blood compatibility.” (Page 1, Line 18-19)

2 CommentIntroduction: Lines 56 and 59, please, consider also the revision of the text. Specially, line 59 when talking about "cascade of reactions". For to accomplish coagulation. Are there some reactions involved?? and which are the type of reactions?? When you check results, you take, only, into account the possibility of coagulation, but the presence of NH2 groups on surface (as you demonstrated clearly) will allow some secondary reactions

Response: This cascade reactions mean the coagulation cascades and inflammatory cascades in biological. Our statement may easy to confuse with the cascade reaction in chemical. The expression has been revised. (Page 2, Line 74-76)

3 CommentLine 141. You refer to a specific structure of polyimide that is the same as "according to literature". I've searched on internet and there are several possibilities for this structure. It would be interessting to add some bibliographic references to make easy to understand the trascendency of this "structure".

Response: The polyimide structure refers to the article of “Analysis of plasma polymerization of allylamine by FTIR” (Krishnamurthy, V.; Kamel, I.L.; Wei, Y. Journal of Polymer Science Part A: Polymer Chemistry 1989, 27, 1211-1224). This literature has been supplied to the references (references 29).

4 CommentLine 161: You attribute several physical changes on the surface of HT-PPA's to the loss of colatile compounds or low molecular weight systems. This can explain some results, but, in my opinion, in a system that has been activated by plasma treatment, when annealing there should be some cross-linking reactions attending to the chemical structures involved and their tridimensional orientation.That fact should promote a light contraction that would explain holes and stripes that can be seen on Fig2 and Fig 6. Those bonds can not be detected using ATR.FTIR

Response: Thanks for the Reviewer’s suggestions. The cross-linking reactions in the annealing process should be taken into account. The annealing treatment could cause the further crosslinking in the PPAa film, and that increased the stability of the polyalimine film. Meanwhile, the annealing treatment also might cause a light contraction. So, the thickness and roughness of HT-PPAa film was lower than PPAa, and a slight stripe was also observed by the AFM. While, the holes and stripes shown in Fig 6 were observed after soaking. They may cause by the water absorption and partial cross-linking of the coating (you mentioned after). We had revised the manuscript and discussed the cross-linking reactions during the annealing process. (Page 5, Line 176-179)

5 CommentLine 164. In my opinion, and from data in graph on Fig 5, density of PPAa should be more 54 or 53 than 45.3 nmol/cm2. The expression used "reduced" used after, confirms my impression.

Response: We are very sorry about this big mistake. The Surface -NH2 density of PPAa was 53.6 nmol/cm2 and the Surface -NH2 density of HT- PPAa was 45.3 nmol/cm2. It have been corrected in the revised manuscript. (Page 6, Line 196)

6 CommentLine 175-176. Swelling, usually never creates cracks or stripes, because is limited by the transversal interactions between chemical groups, and it is usually limited except in viscoelastic systems. In my opinion, that will be caused by partial cross-linking effects, mentioned before.

Response: It is a valuable suggestion. The main reason caused the cracks and stripes is the partial cross-linking. It may be not properly to use swelling for this phenomenon. Wrinkling should be better. When the coating immersed into the PBS solution, the water could be into the coating. So, the swelling was happened. However, the coating was limited by the substrates. So, the wrinkling was occurred in the weak part of the coating. Indeed, the HT-PPAa coating was partially crosslinked. So, the cracks were observed over 10 days immersing. We introduced this point in the revised manuscript. (Page 7, Line 212-216)

Reviewer 2 Report

I strongly suggest the authors to get the manuscript checked by the native english speaker or use the MDPIs english correction facility. Attached are some of my comments/suggestions.

Author Response

Dear Editors and Reviewers:

Thank you for your letter and the reviewers’ comments concerning our manuscript entitled “An albumin biopassive polyallylamine film with improved blood compatibility for metal devices(polymers-486496). Those comments are all valuable and very helpful for revising and improving our paper, as well as the important guiding to our researches. We have studied reviewer’s comments carefully and made revision which marked in blue in the text. We have tried our best to revise our manuscript according to the comments, which we hope to meet with approval.

The list of revision and reply:

Anti-thrombogenicity is the upmost important factor determining the success of the metal devices which are used for continuous contact with the blood. I commend the authors for addressing this issue. The methodology developed and the conclusions drawn from the results are good. In many stances, sentences are too long, clumsy and hard to understand. Numerous grammatical errors and sentence formation errors were found. I strongly suggest the authors to get English checked with native English speaker or seek help from the MDPI English correction department. Only then I could suggest this paper for acceptance.

Response: Thanks for the reviewer’s kind suggestion. We have been carefully checked the full text for grammatical errors, and we also invited one German friend to help us for English correction.

1 CommentLine 10: Two past tense words cannot be in a same sentence. “employed to made” should be “employed to make”.

Response: Thanks for the Reviewer’s comments. This grammatical error had been corrected. (Page 1, Line 10)

2 CommentLines 12-14: Please paraphrase the sentence for better readability.

Response: The sentence has been revised to “Blood contact devices usually need to have good mechanical properties for keeping the clinical performance. Metal materials are one desirable candidate to fabricate blood contact devices due to their excellent mechanical properties and machinability.” (Page 1, Line 12-15)

3 CommentLine 14-15: Blood contact devices are medical devices themselves. What does the author want to convey here? Comparing the blood contact devices with what? Please be specific.

Response: Thanks for pointing out the complexity of the text here. We want to explain that blood-contact materials have better blood compatibility than other biomaterials, like artificial joints, artificial crystals. The text had been revised in the manuscript. (Page 1, Line 16)

4 CommentLine 20: Units: Please be consistent with the units. In some stances there is a space between number and unit (Line 20: 150 ℃, while some stances do not have a space (Line 92: 150℃). Please be consistent and stick to single formatting type.

Response: In the revised manuscripts, all the units were revised to be consistent, with a space between number and unit.

5 CommentDoes the authors mean, they increased using some methods or is it that they are conveying the results? if it stating the results, it should have been 'stability of HT-PPAa film was more than the PPAa film' or 'HT-PPAa showed increased stability when compared to PPAa film'. Please correct accordingly.

Response: Thanks for the Reviewer’s comments. We were stating the results, so we revised the text. (Page 1, Line 25-26)

6. CommentIntroduction is too short. Please try to add some previous/ongoing research in this field. Mention different approaches followed by the researchers to solve this problem. What are their advantages and disadvantages? How is your research different and advantageous from them?

Response: The relative researches have been added to the introduction. The different and advantageous of this work were also described. (Page 2, Line 58-76)

7. CommentLines 42-44: Sentence is too long and confusing. Please breakdown the sentence appropriately.

Response: The sentence has been revised to “However, none of the blood-contacting materials used in clinical can meet all the hemocompatibility requirements, whether the short-term applicated or the long-term implanted”. (Page 1-2, Line 43-45)

8. CommentLines 72-75: Sentence is too long. Hard to understand. Please breakdown into simpler terms.

Response: The sentence has been revised to “The surface chemical composition was characterized by FTIR, XPS. The surface topography was evaluated by AFM, and the surface wettability was evaluated by wetting angle. (Page 2, Line 92-94)

9. CommentPlease list all the materials used for the study with the appropriate source of acquisition. The reason for having the materials section is to know the exact type of chemical to be used if in case of reproducibility.

Response: The sources of reagents were provided into the metarials and method section of the revised manuscript. (Page 3, Line 103-105)

10. CommentLine92: What are PPam's? Is that a different composition than PPAa? If so, please mention it. or is it just a typo.

Response: We are very sorry for this spelling mistake. Thanks a lot for your correction. It should be PPAa. The text has been revised. (Page 3, Line 113-114)

11. CommentPlease add a section of statistical analysis which includes how many number of samples were been tested for each experiment. What type of statistical analysis were performed to determine the best out of them?

Response: The statistical analysis had been added into the revised manuscript. The sample size and statistical analysis results were also supplied to the figures captions. (Page 4, Line 147-151)

12. CommentConclusions section can also have some discussion about the results you’ve obtained. What are the applications of this research? How is it ground breaking in this field? What are the future works?

Response: Thanks for the kind comment. Some discussion has been supplied to the conclusion section. (Page 10, Line 283-291)

Round  2

Reviewer 2 Report

Thank you for considering the suggestions. I would suggest getting the manuscript checked for grammatical errors one more time thoroughly before its ready for publication.